

# Comparison of Global Datasets of Sodium Densities in the Mesosphere and Lower Thermosphere from GOMOS, SCIAMACHY and OSIRIS Measurements and WACCM Model Simulations from 2008 to 2012

Martin P. Langowski[1], Christian von Savigny[1], John P. Burrows[2], Didier Fussen[3], Erin C. M. Dawkins[4,5], Wuhu Feng[6,7], John M. C. Plane[6], and Daniel R. Marsh[8]

[1]Institute of Physics, Ernst-Moritz-Arndt-University of Greifswald, Greifswald, Germany
[2]Institute of Environmental Physics, University of Bremen, Bremen, Germany
[3]Belgian Institute for Space Aeronomy, Brussels, Belgium
[4]NASA Goddard Space Flight Center, Greenbelt, MD, USA
[5]Department of Physics, Catholic University of America, Washington, DC, USA
[6]School of Chemistry, University of Leeds, Leeds, UK
[7]National Center for Atmospheric Science, School of Earth and Environment, University of Leeds, Leeds, UK
[8]National Center for Atmospheric Research, Boulder, CO, USA

*Correspondence to:* Martin Langowski
(langowskim@uni-greifswald.de)

**Abstract.** During the last decade, multiple limb sounding satellites have measured the global sodium (Na) number densities in the mesosphere and lower thermosphere (MLT). Datasets are now available from GOMOS, SCIAMACHY (both on Envisat) and OSIRIS/Odin. Furthermore, global model simulations of the Na layer in the MLT simulated with WACCM-Na are available. In this paper,

we compare these global datasets. Globally, there is an agreement in the observed and simulated monthly average of Na vertical column densities that were compared with each other. They show a clear seasonal cycle with a summer minimum most pronounced at the poles. They also show signs of a semi-annual oscillation in the equatorial region. The vertical column densities vary between $0.5 \times 10^9 \, \mathrm{cm}^{-2}$ to $7 \times 10^9 \, \mathrm{cm}^{-2}$ near the poles and between $3 \times 10^9 \, \mathrm{cm}^{-2}$ to $4 \times 10^9 \, \mathrm{cm}^{-2}$ at the

equator. The phase of the seasonal cycle and semi-annual oscillation shows small differences between the different instruments. The full width at half maximum of the profiles is 10 to 16 km for most latitudes, but significantly smaller in the polar summer. The centroid altitudes of the measured sodium profiles range from 89 to 95 km, while the model shows on average 2 to 4 km lower centroid altitudes. This coincides with a 3 km lower mesopause altitude in the WACCM simulations com-

pared to measurements, which may be the reason for the low centroid altitudes. Despite this global 2 to 4 km shift, the model captures latitudinal and temporal variations. The variation of the WACCM dataset during the year at different latitudes is similar to the one of the measurements. Furthermore, the differences between the measured profiles with different instruments and therefore different lo-



cal times are also present in the model simulated profiles. This capturing of latitutinal and temporal
variations is also found for the vertical column densities and profile widths.

## 1   Introduction

The metal layers in the mesosphere and lower thermosphere (MLT) are formed by ablation from
meteoroids entering the Earth's atmosphere (see, e.g., Plane, 2003 and Plane et al., 2015 for reviews).
The main source of these meteoroides is cometary dust from the Jupiter-family comets (see, e.g.,
Nesvorný et al., 2010), which produce a dominating continuous input. The Jupiter-family comets
have orbits with periods of less than 20 years. Their current orbits are dominated by the gravitational
field of Jupiter and are contained within or do not extend much beyond the orbit of Jupiter (see,
e,g, Levison, 1996 for a classification of comets). Additionally, the Earth passes comet trails of
sublimating short-period comets, orbiting the sun with typical periods of around 100 years which
cause meteor showers at certain periods during the year. This highly varying input, however, does
not significantly increase the densities of the metal layers (see, e.g., Correira et al., 2010). The
meteoroids that enter the Earth's atmosphere have geocentric speeds between 11.5 to 72.5 km s$^{-1}$
and a mass distribution between $10^{-10}$ and $10^{-1}$g, with current estimations from Nesvorný et al.
(2010), Love and Brownlee (1993) and Fentzke and Janches (2008) showing a maximum on the
order of magnitude of several $\mu$g to several hundreds of $\mu$g (see, Carillo-Sánchez et al., 2015 for a
comparison and detailed discussion on this issue). The ablation process (see, e.g., McNeil et al., 1998
and Vondrak et al., 2008) takes place at altitudes between 80 to 125 km, resulting in the deposition
of metallic atoms such as sodium (Na), magnesium (Mg), iron (Fe), potassium (K), calcium (Ca),
nickel (Ni) and others in the MLT. At the upper edge of the metal layers the metal atoms are ionized.
Throughout the whole layer, especially at the bottom, the metals react into molecular species like
carbonates, hydroxides and oxides (see, e.g., Plane et al., 2015). These molecules further react
into so called meteoric smoke particles (see, e.g., Hunten et al., 1980, Kalashnikova et al., 2000
and Saunders and Plane, 2006). These meteoric smoke particles are thought to play a significant
role in the formation of noctilucent clouds (see, e.g., Rapp and Thomas, 2006) in the summer polar
mesosphere and for aerosols and clouds in the stratosphere (see, e.g., Voigt et al., 2005 and Curtius
et al., 2005). However, to quantify the impact of meteoric smoke on the middle atmosphere, it is
important to understand the changes in chemical composition of the incoming particles during entry
into the Earth's atmosphere (e.g., Rudraswami et al., 2016) and how much meteoric material is on
average deposited into the Earth's atmosphere. The rate of daily influx of meteoric material into the
upper atmosphere has a large uncertainty with estimates varying between 1 to 300 tons per day (see,
e.g., Table 1 of Plane, 2012).

   The metal layers in the MLT have been first observed by Slipher (1929) (who could not prove
whether the sodium is from the Earth's atmosphere or from space) by the means of photometry. To





date, in situ measurements of the metal layers are relatively sparse. The reason for this is that bal-
loons are only able to fly up to 50 km altitude, and the atmospheric drag on satellites is too strong
for stable satellite orbits in the altitude of the metal layers. Therefore, in situ measurements are only
possible with rockets, which are relatively expensive compared with other measurement methods
and additionally can only be deployed at very few locations on Earth. In situ rocket mass spectrom-
eter measurements of metal ions have first been reported by Johnson and Meadows (1955). Until
2002, approximately 50 flights of rocket borne mass spectrometers probing the MLT region had
occurred according to Grebowsky and Aikin (2002). Due to this lack of in situ measurements, the
investigation of the mesospheric metal layers heavily relies on remote sensing methods. Quantitative
'ground-based' observations have been made since the 1950s with photometers measuring resonance
fluorescence radiation of the metal atoms that scatter the solar radiation. On the ground, photome-
ters were superseded in the 1970s by the lidar technique (light detection and ranging) which provides
several advantages: Lidar makes it possible to measure at any time of the day, whereas photometry
only operates at twilight. In the context of the metal emission lines, the Sun is not an ideal light
source as its spectrum usually has a minimum of spectral radiance (formally known as Fraunhofer
lines) at the metal spectral lines with a spectral structure that needs to be measured at sub-pm res-
olution. This shows significant Doppler-shifts and varies with time (especially strong for $Mg^+$). In
contrast to that, the lasers have a maximum intensity at the desired wavelengths and a well-known
spectrum. The intensity at a certain wavelength that is needed for a good signal to background ratio
can be achieved by using the appropriate laser. Thus it is possible to measure metal densities not
just for the ground state but also for different excited states and from this temperatures can also be
derived. An overview of the locations of recent 'ground-based' lidar measurements is given by Plane
et al. (2015) (their Fig. 11). 'Ground-based' lidar measurements provide metal density profiles with
very good vertical and temporal resolution but are stationary and limited to singular points on Earth.
Thus, global coverage can only be achieved by a large network of ground stations or the use of a
mobile basis like a satellite.

Only in the last decades have 'space-borne' spectrometer measurements provided number den-
sity profiles or column density datasets with (nearly) global coverage for continuous time periods of
several years. These 'space-borne' spectrometers typically were 'on-board' of satellites with sun-
synchronous and polar orbits and a maximum scanned latitude of up to 82 degrees, that retrieved
densities for Mg and/or $Mg^+$ (see, e.g., Joiner and Aikin, 1996, Correira et al., 2008, Scharring-
hausen et al., 2008 and Langowski et al., 2015), K (see, e.g., Dawkins et al., 2014) and Na (see, e.g.,
Fussen et al., 2004, Casadio et al., 2007, Fan et al., 2007, Gumbel et al., 2007, Fussen et al., 2010,
Hedin and Gumbel, 2011 and Langowski et al., 2016). Along with the measurements, global atmo-
spheric models for the metal layers in the MLT have been developed for Na (Marsh et al., 2013a),
Fe (Feng et al., 2013), Mg (Langowski et al., 2015), K (Plane et al., 2014, Feng et al., 2015 and
Dawkins et al., 2015), and Si (Plane et al., 2016) atoms, molecules and ions.

The global datasets for Na appear to be similar but a direct comparison of these datasets has not been carried out thus far. In this study, we compare the latest global datasets for Na obtained from GOMOS/Envisat, SCIAMACHY/Envisat, and OSIRIS/Odin measurements along with the WACCM-Na model results on a global level. We check where the measurements need improve-
ment in accuracy and how good the model captures the observations as well as where the model needs further improvements. In Sect. 2 the instruments/model from which the Na densities are re-trieved are described and an error estimation for the measurements is presented. For all four datasets the Na densities are available for different latitudes, altitudes and times. In this paper we focus on the key profile characteristics of the Na layer, which includes the vertical column densities (VCDs)
(Sect. 3), the centroid altitude of the profile (Sect. 4), as well as the profile width in the form of the Full Width at Half Maximum (FWHM) (Sect. 5). Finally, the key results of this comparison are summarized in Sect. 6.

## 2   Instruments and model information

Before we compare the different datasets in this study, first some basic information on the involved
instruments and techniques are given in this section. However, the focus of this study shall be on the comparison of the datasets, so that the instruments and techniques, which are well documented by Fussen et al. (2010), Gumbel et al. (2007) and Langowski et al. (2016) are only briefly described. In addition, information on the errors of the different datasets are provided, which is useful when comparing different datasets.

### 2.1   GOMOS

In this study we use calculated Na densities using the model formulae given by Fussen et al. (2010). These formulae are derived from fits to the actual GOMOS (Global Ozone Monitoring by Occulta-tion of Stars) measurements during the period of 2002 to 2008. GOMOS 'on-board' the European Space Agency's (ESA) satellite Envisat was launched into space on February, 28, 2002. Envisat flies
on a Sun-synchronous orbit at around 800 km altitude, crossing the equator from north to south at around 10 a.m. local time, and from south to north at around 10 p.m. local time. Between $\pm 60$ degree latitude the local time varies within $\pm$ 1 hour from the equatorial crossing time. For the descending part of the orbit, where the satellite flies from north to south, local time shifts to later hours in the north and earlier hours in the south. One orbit takes approximately 100 minutes, which
corresponds to roughly 14.5 orbits per day. In April 2012, the communication with Envisat broke abruptly and it was not possible to reestablish contact.

Regarding studying the Earth's atmosphere, GOMOS was one of the first instruments to routinely exploit the principle of stellar occultation (see, e.g., Kyrölä et al., 2004 and Bertaux et al., 2004) and allowed the first global measurement of the upper atmospheric Na layer from 2003 (Fussen



et al., 2004). The telescope system connected to the GOMOS spectrometer channels is able to track
stars. The measurement principle is to measure the radiation of a star with and without the Earth's
atmosphere between the star and the instrument, to determine how much radiation is absorbed and
scattered in the Earth's atmosphere. This is done for around 20 to 40 occultation measurement
sequences per orbit, in which a star is followed from a tangent altitude of about 10 km to 150 km, at

daylight and night conditions, which sums up to around $550,000$ star occultations from 2002 to 2008.
The absorption features of the Na D-lines at 589 nm are used to retrieve Na number densities. A
DOAS (Differential Optical Absorption Spectroscopy) technique is used to retrieve slant path optical
thicknesses, from which the Na number densities are derived. Details on the most recent version of
the retrieval algorithm are given by Fussen et al. (2010). In the context of this study it should be

noted that dark limb measurements during night conditions have a larger number of occultations
with a higher statistical significance than the bright limb measurements during daylight and twilight
conditions. This also means, that during the polar summer, where only daylight measurements are
available the statistical significance is lower than for the other latitudes and times.

     The concrete formulae used in this study are Eqs. (8), (9) and (11) along with the parameters

from Table 1 by Fussen et al. (2010), which are repeated here. These formulae consider the most
important variation effects of the Na density field however this also means that not every detail of
the measurements is captured, which results in a smoother density field compared with the actual
measurements. A comparison of the formulae and actual measurements is shown by Fussen et al.
(2010) in their Fig. 9 and 12.

The formula for the VCD $N$ for a certain month $m$ (january is $m = 0$) and latitude $\phi$ (in radians
for Eq. 1 and Eq. 2) is:

$$N(m,\phi)[\text{cm}^{-2}] = t_0 + t_1 \cos\left(\frac{2\pi}{12}m + t_2\right) + t_3\left(\phi + \frac{\pi}{2}\right)\left(\phi - \frac{\pi}{2}\right)\cos\left(\frac{2\pi}{6}m + t_4\right)$$

$$t_{i \geq 1} = f_i\left(a_0 + a_1\phi + a_2\phi^2 + a_3\phi^3\right) \tag{1}$$

$$t_0 = 3.28 \times 10^9$$

The parameters in Eq. (1) are given in Table 1. This formular considers a constant component $t_0$,

| $i$ | $f_i$ | $a_0$ | $a_1$ | $a_2$ | $a_3$ |
|---|---|---|---|---|---|
| 1 | $1 \times 10^9$ | 0.1282 | 1.549 | 0.1780 | 0.03511 |
| 2 | 1 | 0.4017 | 0.8216 | -0.1282 | -0.2980 |
| 3 | $1 \times 10^9$ | -0.2630 | 0.1121 | 0.6355 | -0.3566 |
| 4 | 1 | -1.5635 | -3.0526 | 1.3802 | 1.7637 |

**Table 1.** Parameters for Eq. (1)

a yearly cycle and a semi-annual cycle. The annual cycle is most pronounced in the polar region
and its phase and amplitude are determined by the parameters $t_1$ and $t_2$. The semi-annual cycle,
whose amplitude and phase are determined by the parameters $t_3$ and $t_4$, is most pronounced in the



equatorial region. The different $t_i$ are 'third-order' polynomials in latitude $\phi$. The fit uncertainty is about $\delta N \approx 0.81 \times 10^9\,\mathrm{cm}^{-2}$. The formula for the peak altitude $z_p$ (which is the same as the centroid altitude for a 'Gaussian-shaped' layer) for a certain month $m$ and latitude $\phi$ is:

$$z_p(m,\phi)[\mathrm{km}] = (91.98 - 0.7723\phi^2) + (0.1364 - 0.6532\phi^2)\cos\left(\frac{2\pi}{12}m + 1.302 - 0.887\phi\right) \quad (2)$$

The peak altitude $z_p$ is highest at the equator and on average 2 km lower at the poles. This is overlaid with a seasonal cycle component, which has a 160 degree phase shift between the variation at both poles. On average a standard deviation of 1.6 km is observed for different latitudes and months. The profile width of the Na layer given by Fussen et al. (2010) is not determined for individual latitudes. Instead, one global FWHM is determined as $\mathrm{FWHM} = (12.2 \pm 3.6)\,\mathrm{km}$, which for a 'Gaussian-shaped' profile corresponds to width parameter $\sigma$ of $\sigma = \mathrm{FWHM}/\sqrt{8\ln 2} = (7.3 \pm 2.2)\,\mathrm{km}$.

## 2.2 SCIAMACHY

The SCanning Imaging Absorption spectroMeter for Atmospheric CHartography SCIAMACHY (see, e.g., Burrows et al., 1995 and Bovensmann et al., 1999) is also 'on-board' Envisat which was described in Sect. 2.1. SCIAMACHY has different measurement geometries, of which the limb MLT measurements were used to retrieve Na densities from resonance fluorescence of the Na D-lines at 589 nm wavelength. The radiation source to trigger the resonance fluorescence is the sun, so that only the sunlit part of the orbit can be observed with this method. However, a method to retrieve Na from the SCIAMACHY nightglow measurements has recently been developed by von Savigny et al. (2016) but it is still in a preliminary phase and is not considered in this study.

Na densities were retrieved from both D-lines and the arithmetic average of both is used in this study. The limb MLT measurements of SCIAMACHY were performed roughly every two weeks from 2008 to 2012 for 15 consecutive orbits, which corresponds to roughly one day of consecutive limb MLT scans. This resulted in 83 days of limb MLT measurements which were used for the Na number density retrieval. Na number densities were retrieved from daily zonally averaged data and from this multiannual averages for each month were formed. Each limb MLT scan consists of 30 limb measurements with tangent altitudes between 50 and 150 km and with a step size of around 3.3 km. A finer grid with 1 km stepsize was used in the retrieval algorithm, because of numerical reasons (for example when calculating the Beer-Lambert law for absorption, the exponential formula is only evaluated at discrete points, which quasi transforms the exponential formula into a stepwise linear function, which is closer to the true exponential expression the smaller the steps are). However, the vertical resolution is mostly determined by the vertical sampling of the tangent altitudes and is very close to the 3.3 km step size. The retrieval grid uses 40 latitudes bins between 82° N and S for the descending part of the orbit which corresponds to a latitudinal sampling of roughly 4.1 degree latitude. More details on the retrieval of the SCIAMACHY dayglow Na dataset are described by Langowski et al. (2016).





The statistical error of the vertical profiles is roughly $10\%$ in the peak altitude and is similar for the VCD. However, as Na is retrieved independently from both D-lines, both individual results can be compared, which was done in Langowski et al. (2016). For most latitudes and months the relative differences between the Na $D_1$ and Na $D_2$ line results are within $\pm 10\%$. However, the differences are larger at the highest latitudes during the southern hemispheric winter, with absolute differences of the VCDs of up to $3 \times 10^9\,\mathrm{cm}^{-2}$, which corresponds to a relative difference of $40\%$. For this study, we use the arithmetic mean of the densities from the $D_1$ and the $D_2$. With respect to the differences this means, that the difference of the mean to the two individual density fields is half as large as the difference between the two individual density fields. Errors for the centroid altitude and FWHM are not provided by Langowski et al. (2016), but are estimated to be well $< 1\,\mathrm{km}$. One systematic error source when determining the centroid altitude is an error in the determination of the tangent altitude of the used measurements. Bramstedt et al. (2012) showed that the tangent altitude information used in this study is accurate within a few hundred meters, which is a big improvement compared to the initial phase of the SCIAMACHY mission which had errors of up to 5 km (von Savigny et al., 2005).

### 2.3 OSIRIS

The Optical Spectrograph and InfraRed Imager System (OSIRIS, see e.g., Llewellyn et al., 2004) is one of two instruments located 'on-board' the Odin satellite. Launched on a START-1 rocket on February, 20, 2001 from Svobodny, Russia, Odin is a still operational, dual-purpose astronomy and aeronomy mission, designed and managed by a Swedish, Canadian, Finnish and French consortium. The Odin satellite flies at approximately 600 km altitude in a sun-synchronous, polar orbit with an inclination angle of $97.8°$, resulting in coverage extending between $82°$N to $82°$S. Completing approximately 15 orbits per day, the satellite has two local equator-crossing times at 0600 LT and 1800 LT on the descending and ascending nodes, respectively. Due to orbital drift, over time these equator-crossing times have become progressively later and are now closer to 0650 LT and 1850 LT. The OSIRIS instrument measures limb-scattered sunlight across the wavelength range 280-810 nm, with a pixel resolution of 0.4 nm and spectral resolution of 1 nm. The satellite performs limb scans between 5 to 110 km, with a typical height resolution of 1.5-2 km within the mesosphere and the instrument field-of-view is approximately 1 km vertically and 40 km horizontally, when mapped onto the atmospheric limb at the tangent point. As the observation of solar induced resonance fluorescence relies on daylight conditions, there is limited coverage during the winter hemisphere's polar night at middle to high latitudes. The OSIRIS Na retrieval scheme was developed by Gumbel et al. (2007) and is an optimal estimation method after Rodgers (2000) which uses a forward model to convert OSIRIS-observed limb radiances of the Na D-line resonance scattering at 589 nm into vertically resolved Na number densities. The observed spectra are modeled by integrating the radiation scattered toward the instrument along the line-of-sight in a spherical atmosphere, with background temperature and density profiles taken from the Mass Spectrometer Incoherent Scatter atmospheric model



(see, e.g., Hedin, 1991) and European Centre for Medium-Range Weather Forecasts (ECMWF ERA-
Interim) reanalyses (see, e.g., Dee et al., 2011). The OSIRIS Na dataset consists of vertical number
density profiles between 75-110 km, with a vertical resolution of 2 km and a typical uncertainty of
10%.

## 2.4 WACCM-Na

For this study we simulated the Na species during the period of 2008-2012 using an updated version
of WACCM-Na which was originally developed by Marsh et al. (2013a). In the study we used the
version 4 of the Whole Atmosphere Community Climate Model (WACCM, see, e.g., Marsh et al.,
2013b) with the inclusion of the Na chemistry (see Marsh et al., 2013a) and a few updated reactions
based on the recent work in Plane et al. (2015) and Gómez Martín et al. (2016) under the numerical
framework of NCAR Community Earth System Model, version 1 (CESM1, see, e.g., Hurrell et al.,
2013). WACCM is a high-top coupled chemistry-climate model with an upper boundary at $6.0 \times
10^{-6}$ hPa, which corresponds to an altitude of $\approx 140$ km and integrates atmospheric chemistry and
physics from the troposphere up to lower thermosphere with a detailed description of mesospheric
and lower thermosphere processes (see, e.g., Marsh et al., 2007) as well as detailed formulations
of radiation, planetary boundary layer turbulence, cloud microphysics and aerosols (see, e.g., Mills
et al., 2016). The model horizontal resolution is $1.9° \times 2.5°$, with a vertical resolution in the MLT of
less than 500 m which is identical as in Viehl et al. (2016) by increasing the hybrid sigma-pressure
vertical coordinate from 88 to 144 levels, using the same method as Merkel et al. (2009). WACCM is
nudged with specified dynamics using meteorological fields from the NASA Global Modelling and
Assimilation Office Modern-Era Retrospective Analysis for Research and Applications (termed as
MERRA, see, e.g., Rienecker et al., 2011) below 60 km. The Prandtl number was set to 2 here which
is suggested by other MLT studies, e.g., Garcia et al. (2016). The meteoric input function for Na is
described in Marsh et al. (2013a). WACCM-Na, in the following just called WACCM, satisfactorily
reproduces the seasonal cycle of the Na layer (column density, peak concentration, layer height, and
top- and bottom scale heights) when compared with satellite and lidar observations (see, e.g., Marsh
et al., 2013a and Dunker et al., 2015). The modelled global fields are saved at 0 universal time every
day during the simulation period.

## 2.5 Homogenization of the datasets for comparison

As the different datasets cover different time, latitude and altitude ranges, the datasets have to be
colocated and interpolated. The WACCM local time is colocated to the different satellite local times
by using global output at 0:00 UTC. The different longitudes then correspond to different local
times and only the data within ± 1 hour of the satellite times is used to average for the latitude
bins. These averages are assumed to be representative for zonal averages. This local time colo-
cation with the different satellite experiments is shown in Fig. 1. As the model and measurement




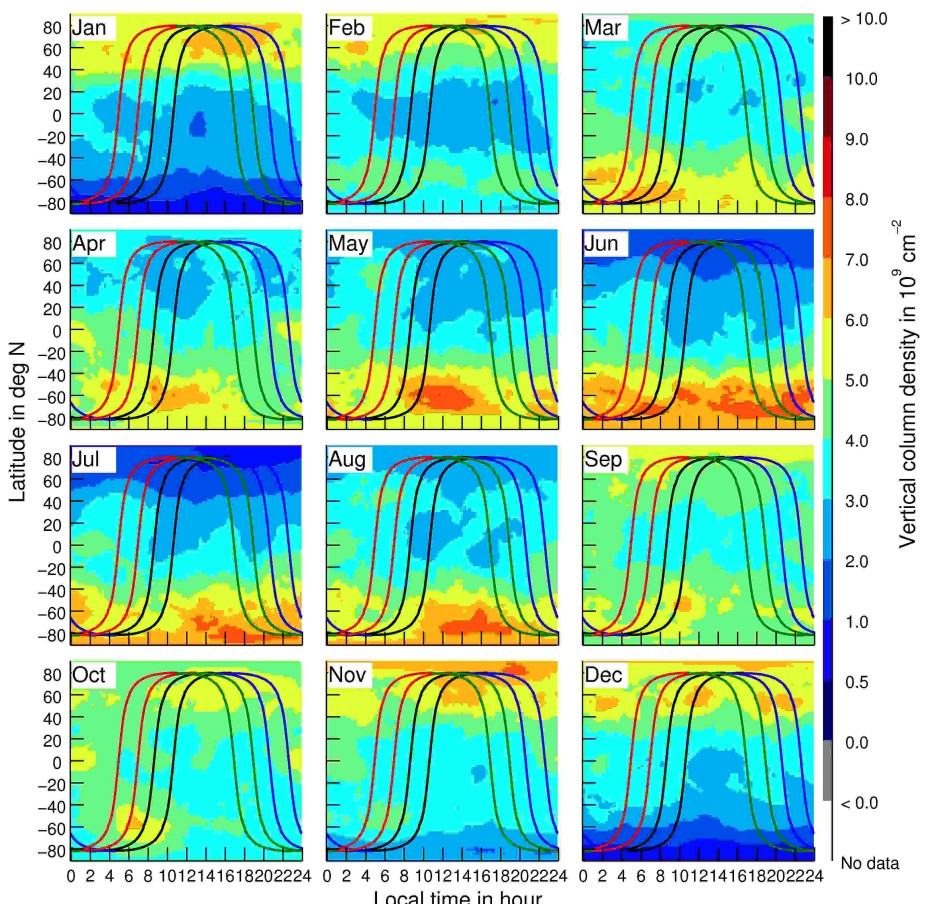

**Fig. 1.** WACCM vertical column densities of multiannual monthly means 2008-2012 for different local times. The satellite data is colocated ± 1 hour, which is indicated by the blue lines for the ascending leg of GOMOS, by the red lines for the descending leg of OSIRIS, by the green lines for the ascending leg of OSIRIS and by the black lines for the SCIAMACHY dayglow measurements as well as for the GOMOS descending leg.

results are calculated for different latitude and altitude grids, the data is colocated to the instruments with the coarsest grid resolution. The degrading of the resolution of the better resolved dataset to the resolution of the dataset with the coarsest resolution is done by forming weighted means. For example, for the coarse altitude interval at 76 km which spans from 75 km to 77 km, while the altitude interval of the finer SCIAMACHY grid is 1 km, the following weighting formula is used: $n_{\text{coarse}}(76\,\text{km}) = (0.5 n_{\text{fine}}(75\,\text{km}) + n_{\text{fine}}(76\,\text{km}) + 0.5 n_{\text{fine}}(77\,\text{km}))/2$. As the 2 km sampling is still finer than the resolution of SCIAMACHY, which is slightly above 3.3 km, the resolution is not decreased due to the averaging, which also applies to the other data. As the SCIAMACHY Na dataset only includes 83 individual days from 2008 to 2016, data from the same days have been used from



the WACCM and OSIRIS datasets, to form multiannual monthly means. This also means that due
to the data reduction, less data for WACCM and OSIRIS is used for the monthly means (2 days per
month instead of 30) than actually is available, so that both datasets are a bit noisier in this paper's
figures than they actually are. This especially applies to the 'near-terminator' region for OSIRIS,
where sometimes only 1 to 4 individual profiles are used for the averaging, which explains some
outliers. The GOMOS dataset is calculated on the common altitude and latitude grid through the
formulae in Sect. 2.1.

## 3 Vertical column densities and differences

Figure 2 shows the Na VCDs for the different instrument and models. The VCDs are taken for the
altitudes from 76 km to 106 km. Na densities outside of this altitude region are negligibly small.
Figure 3 shows the absolute differences to the ensemble mean VCD and Fig. 4 shows the relative
differences to the ensemble mean VCD. Note that the upper left panels in Figs. 3 and 4 show the
ensemble mean itself with the color bar as in Fig. 2, so that it easier for the reader to see the ensemble
mean and the errors at the same time with the order of the panels being same for all figures.

For the formation of the ensemble mean, first the arithmetic mean of the 4 WACCM-Na density
fields for the different local times is formed. Then, the arithmetic mean of the WACCM-Na mean
and the density fields from the GOMOS measurements, SCIAMACHY measurements and both the
OSIRIS descending and ascending leg measurements is formed. If no instrument data is available at
a certain latitude and time, this instrument is excluded for the averaging at this latitude and time. In
the ensemble mean VCDs range from $0.5 \times 10^9\,\mathrm{cm}^{-2}$ to $7 \times 10^9\,\mathrm{cm}^{-2}$ near the poles and between
$3 \times 10^9\,\mathrm{cm}^{-2}$ to $4 \times 10^9\,\mathrm{cm}^{-2}$ at the equator. The sole purpose of this ensemble mean is to have
a reference dataset to compare with. It doesn't consider a sophisticated weighting of the compared
datasets. It is not better than the individual datasets and some features, e.g., the local time fixation of
the initial datasets are lost due to the averaging. Despite these caveats the ensemble mean in numbers
is given in Table 2 for an easy reproduction as a reference dataset.

Overall, there is a good qualitative agreement between the different datasets; they all show a sea-
sonal cycle with the largest amplitude in the polar region and a polar summer minimum. Also, in the
equatorial region a semi-annual oscillation with maxima in spring and autumn is found in most of the
datasets. When taking a closer look at the differences between the datasets some measurement/model
specific differences can be found. The GOMOS Na VCD is shifted at least a month forward in the
year at nearly all latitudes, which leads to relatively large absolute differences even though the over-
all seasonal cycle is very similar to the ensemble mean. The SCIAMACHY and OSIRIS results
show their largest differences to the ensemble within the 'near terminator' regions. SCIAMACHY
also shows more pronounced discrepancies in the southern hemispheric winter, which is also the
region in which the discrepancies of the separate retrieval of Na densities from the $D_1$ and $D_2$ lines





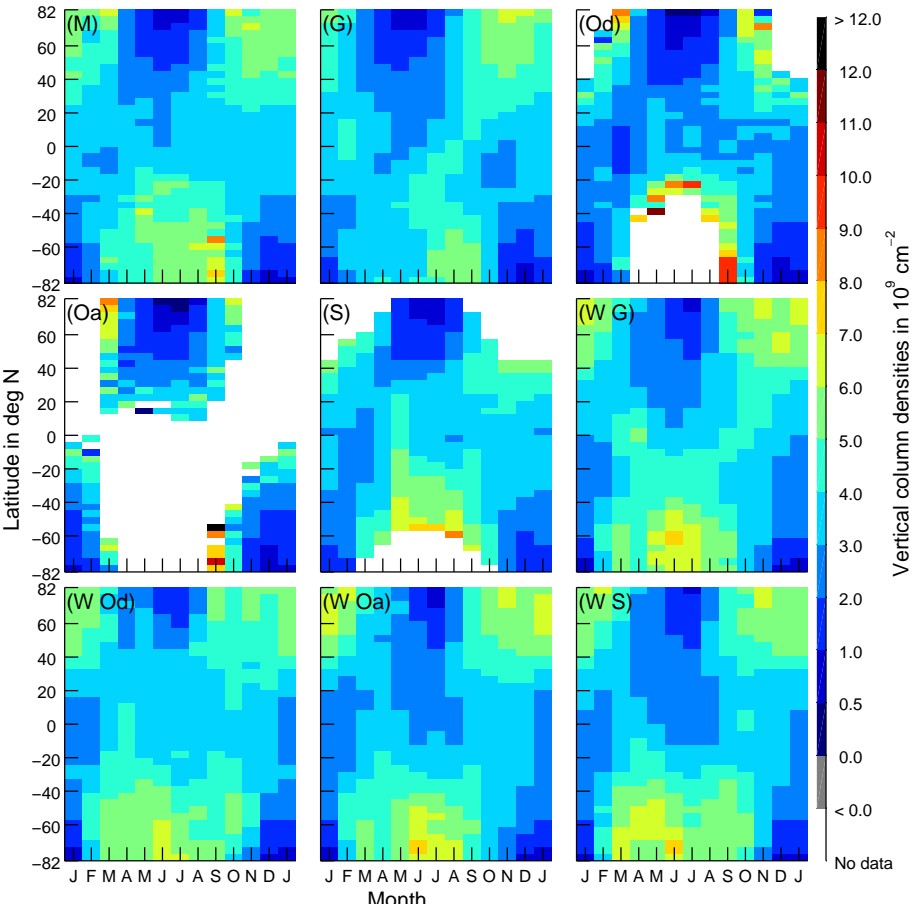

**Fig. 2.** 2008-2012 multiannual monthly mean vertical Na column densities between 76 km and 106 km altitude from different instruments and models. (M) ensemble mean, (G) GOMOS, (Od) OSIRIS descending leg, (Oa) OSIRIS ascending leg, (S) SCIAMACHY dayglow, (W G) WACCM colocated to the local time of ascending leg of GOMOS, (W Od) WACCM colocated to the local time of the descending leg of OSIRIS, (W Oa) WACCM colocated to the local time of the ascending leg of OSIRIS, (W S) WACCM colocated to the dayglow measurements of SCIAMACHY and the descending leg of GOMOS.

from the SCIAMACHY measurements are largest. SCIAMACHY also shows larger vertical column

densities in the equatorial region in May, which is also present in the OSIRIS descending leg results and the corresponding WACCM VCD field. However, this is missing in the GOMOS model data, which is not colocated for the individual days and year so that this feature appears to be a seasonal speciality of the sampled days used, rather than a feature that occurs every year. In the polar summer, the satellite measurements show a slightly stronger decrease in the VCD than the corresponding

WACCM measurements.





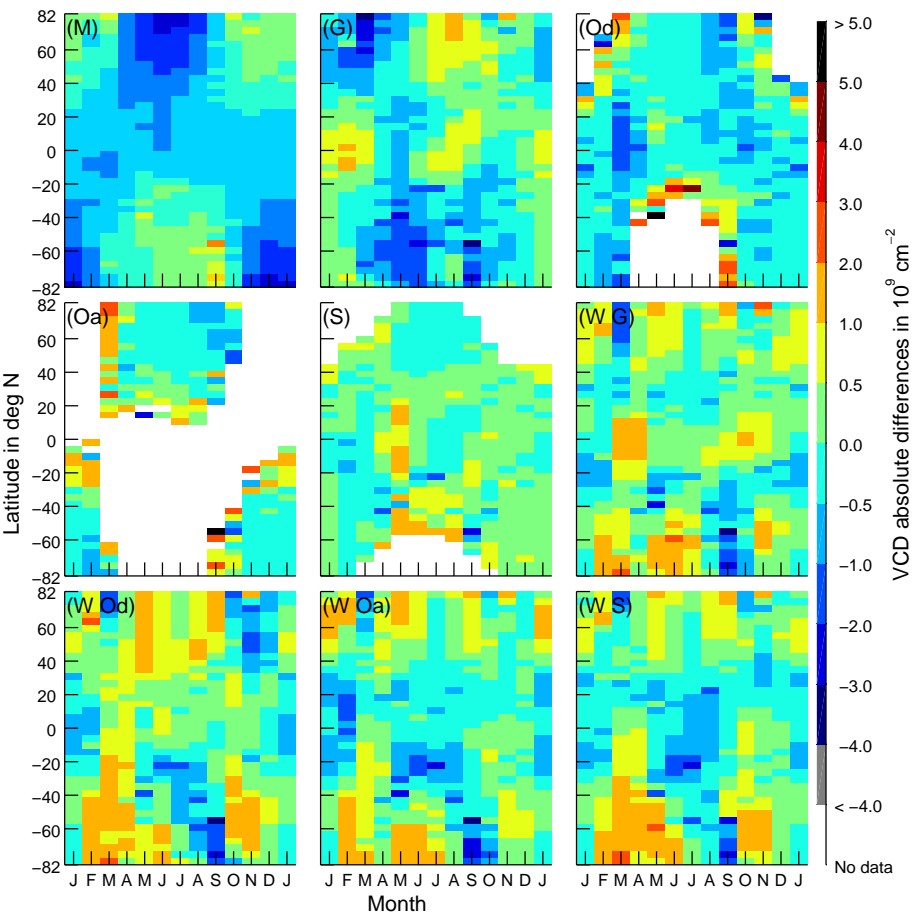

**Fig. 3.** Absolute vertical column density differences of the individual instruments/model results to the ensemble mean. The panels are for the same results as in Fig. 2. The upper left panel shows the ensemble mean with the same color bar as in Fig. 2.

## 4 Centroid altitudes and differences

Figure 5 shows the centroid altitude of the Na layer for the different instruments, and Fig. 6 shows the differences of the centroid altitudes to the ensemble mean, except the upper left panel which shows the ensemble mean in the same color bar as for Fig. 5. This is for the same reason as discussed in

Sect. 3. The centroid altitude retrieved from the satellite measurements range from 89 km up to 95 km, while the Na centroid altitudes derived from WACCM range from 86 km to 92 km and are on average about 2 to 4 km lower than the measured ones. This discrepancy was already discussed by Marsh et al. (2013a) and is most likely attributed to the Mesopause also being about 3 km lower in the WACCM simulations than in satellite observations, showing a strong dependency of the Na



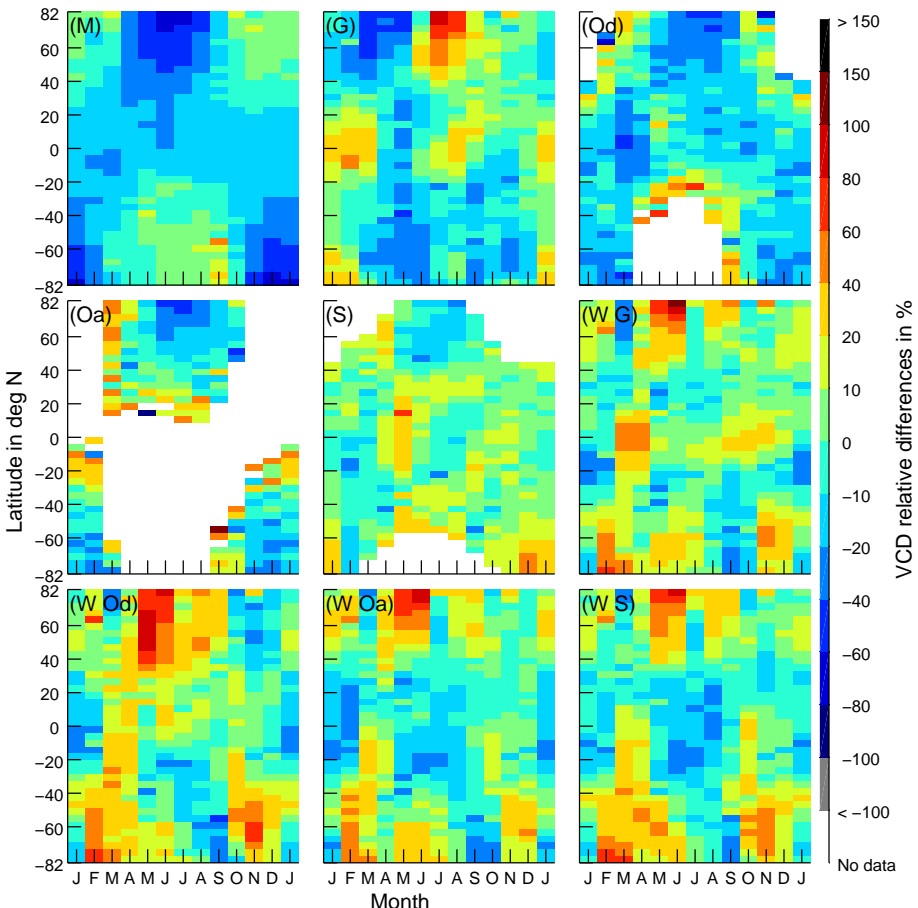

**Fig. 4.** Relative vertical column density differences of the individual instruments/model results to the ensemble mean. The panels are for the same results as in Fig. 2. The upper left panel shows the ensemble mean with the same color bar as in Fig. 2.

layer altitude from the thermal structure in its altitude. For the high latitudes, the centroid altitudes

for all experimental datasets and also in the WACCM results are up to 4 km higher in the summer

than in the winter, respectively, the start and end of the measurement period for the satellites with

no winter coverage. While the centroid altitudes of WACCM are systematically lower, the seasonal

and local time variations of the measurements appear to be well reproduced by the model. For

example, in the low latitudes the profiles from descending leg measurements with OSIRIS have a

higher centroid altitude than the SCIAMACHY profiles, which is a local time effect that appears also

in the WACCM data. For the SCIAMACHY measurements during the summer, there is a minimum

in centroid altitude at mid latitudes while the altitude is higher at the equator and the summer pole,





| Latitude in deg N | Jan | Feb | Mar | Apr | May | Jun | Jul | Aug | Sep | Oct | Nov | Dec |
|---|---|---|---|---|---|---|---|---|---|---|---|---|
| 79.4625 | 5.14 | 5.14 | 6.03 | 2.61 | 1.54 | 0.68 | 0.74 | 1.63 | 3.80 | 6.03 | 4.65 | 5.80 |
| 75.3875 | 4.91 | 4.91 | 5.49 | 2.45 | 1.58 | 0.83 | 0.92 | 1.86 | 3.87 | 5.61 | 5.59 | 5.56 |
| 71.3125 | 4.68 | 4.68 | 5.04 | 2.81 | 1.70 | 0.96 | 1.04 | 1.98 | 3.85 | 5.58 | 6.67 | 5.49 |
| 67.2375 | 4.00 | 4.00 | 4.59 | 2.62 | 1.73 | 1.10 | 1.17 | 2.14 | 3.85 | 5.59 | 5.73 | 5.44 |
| 63.1625 | 3.21 | 3.21 | 4.46 | 2.67 | 1.81 | 1.24 | 1.35 | 2.34 | 4.06 | 4.92 | 5.71 | 5.41 |
| 59.0875 | 4.64 | 4.64 | 4.45 | 2.52 | 1.88 | 1.41 | 1.49 | 2.48 | 3.88 | 4.91 | 5.72 | 5.46 |
| 55.0125 | 4.58 | 4.58 | 4.03 | 2.65 | 1.97 | 1.54 | 1.66 | 2.60 | 3.82 | 4.92 | 5.90 | 5.42 |
| 50.9375 | 4.59 | 4.59 | 4.03 | 2.36 | 2.06 | 1.68 | 1.85 | 2.73 | 3.94 | 4.23 | 5.64 | 5.31 |
| 46.8625 | 4.18 | 4.18 | 3.49 | 2.46 | 2.11 | 1.91 | 2.04 | 2.96 | 4.00 | 4.35 | 5.33 | 5.15 |
| 42.7875 | 4.12 | 4.12 | 3.61 | 2.45 | 2.24 | 2.11 | 2.28 | 3.10 | 3.98 | 4.49 | 5.14 | 4.66 |
| 38.7125 | 4.01 | 4.01 | 3.34 | 2.61 | 2.31 | 2.36 | 2.55 | 3.37 | 4.33 | 4.40 | 4.80 | 4.99 |
| 34.6375 | 3.87 | 3.87 | 3.68 | 2.59 | 2.50 | 2.51 | 2.76 | 3.29 | 3.76 | 4.24 | 4.72 | 4.64 |
| 30.5625 | 3.50 | 3.50 | 3.04 | 2.92 | 2.88 | 2.70 | 3.02 | 3.26 | 4.04 | 4.10 | 4.37 | 4.19 |
| 26.4875 | 3.47 | 3.47 | 3.60 | 2.85 | 3.32 | 2.90 | 3.05 | 3.36 | 3.68 | 4.01 | 4.40 | 4.05 |
| 22.4125 | 3.41 | 3.41 | 3.22 | 3.06 | 3.19 | 2.97 | 3.02 | 3.76 | 3.57 | 3.88 | 4.05 | 3.84 |
| 18.3375 | 3.44 | 3.44 | 3.23 | 3.64 | 3.29 | 2.90 | 3.28 | 3.50 | 3.50 | 3.72 | 3.75 | 3.68 |
| 14.2625 | 3.36 | 3.36 | 3.42 | 3.22 | 2.68 | 3.00 | 3.20 | 3.28 | 3.45 | 3.68 | 3.58 | 3.42 |
| 10.1875 | 3.25 | 3.25 | 3.06 | 3.33 | 3.41 | 2.85 | 3.48 | 3.23 | 3.35 | 3.68 | 3.43 | 3.52 |
| 6.1125 | 3.19 | 3.19 | 3.00 | 3.32 | 3.39 | 2.87 | 3.14 | 3.17 | 3.43 | 3.65 | 3.26 | 3.29 |
| 2.0375 | 3.14 | 3.14 | 3.02 | 3.28 | 3.35 | 2.96 | 3.22 | 3.27 | 3.57 | 3.39 | 3.17 | 3.13 |
| -2.0375 | 3.30 | 3.30 | 2.99 | 3.36 | 3.35 | 3.11 | 3.26 | 3.31 | 3.54 | 3.30 | 3.19 | 3.01 |
| -6.1125 | 2.94 | 2.94 | 2.97 | 3.37 | 3.49 | 3.31 | 3.42 | 3.36 | 3.48 | 3.26 | 3.21 | 3.04 |
| -10.1875 | 2.71 | 2.71 | 2.96 | 3.34 | 3.85 | 3.63 | 3.60 | 3.50 | 3.64 | 3.38 | 3.27 | 3.09 |
| -14.2625 | 3.53 | 3.53 | 2.92 | 3.47 | 4.07 | 3.79 | 3.98 | 3.67 | 3.71 | 3.27 | 3.30 | 3.62 |
| -18.3375 | 3.28 | 3.28 | 3.02 | 3.57 | 4.19 | 4.36 | 4.31 | 3.90 | 3.85 | 3.32 | 3.84 | 3.15 |
| -22.4125 | 3.17 | 3.17 | 3.09 | 3.62 | 4.67 | 5.34 | 5.65 | 4.18 | 4.08 | 3.40 | 3.34 | 3.21 |
| -26.4875 | 3.14 | 3.14 | 3.15 | 3.74 | 4.88 | 5.15 | 4.65 | 4.20 | 4.27 | 3.47 | 3.23 | 3.44 |
| -30.5625 | 2.97 | 2.97 | 3.32 | 4.10 | 5.51 | 4.86 | 4.64 | 4.58 | 4.44 | 3.45 | 3.58 | 2.94 |
| -34.6375 | 3.10 | 3.10 | 3.43 | 4.00 | 4.68 | 4.85 | 4.66 | 4.48 | 4.55 | 3.55 | 3.28 | 2.76 |
| -38.7125 | 3.12 | 3.12 | 3.59 | 4.01 | 6.75 | 4.94 | 4.78 | 5.09 | 4.62 | 3.52 | 3.09 | 2.61 |
| -42.7875 | 3.02 | 3.02 | 3.74 | 4.95 | 5.16 | 4.97 | 4.94 | 5.54 | 4.74 | 4.12 | 3.03 | 2.40 |
| -46.8625 | 3.16 | 3.16 | 3.98 | 4.37 | 5.21 | 5.12 | 5.04 | 5.28 | 5.05 | 3.83 | 2.88 | 2.17 |
| -50.9375 | 3.08 | 3.08 | 4.11 | 4.45 | 5.12 | 5.57 | 5.28 | 5.53 | 5.00 | 3.66 | 2.78 | 2.01 |
| -55.0125 | 3.19 | 3.19 | 4.20 | 4.58 | 5.36 | 5.94 | 5.82 | 5.90 | 8.08 | 3.56 | 2.66 | 1.82 |
| -59.0875 | 2.99 | 2.99 | 4.36 | 4.60 | 4.65 | 5.11 | 5.34 | 6.41 | 6.13 | 4.08 | 2.46 | 1.65 |
| -63.1625 | 2.94 | 2.94 | 4.65 | 4.85 | 4.56 | 5.18 | 5.46 | 5.64 | 5.94 | 3.96 | 2.25 | 1.42 |
| -67.2375 | 2.78 | 2.78 | 5.01 | 4.40 | 4.46 | 5.20 | 5.53 | 5.71 | 6.85 | 4.19 | 2.09 | 1.21 |
| -71.3125 | 2.68 | 2.68 | 4.46 | 4.43 | 4.53 | 5.35 | 5.53 | 5.77 | 6.86 | 4.26 | 2.05 | 1.04 |
| -75.3875 | 2.34 | 2.34 | 3.97 | 4.45 | 4.69 | 5.45 | 5.50 | 5.80 | 7.46 | 4.19 | 1.96 | 0.91 |
| -79.4625 | 2.08 | 2.08 | 3.58 | 4.53 | 4.90 | 5.39 | 5.54 | 5.75 | 6.74 | 4.18 | 1.80 | 0.78 |

**Table 2.** Ensemble mean VCD for 40 latitudes between $\pm 82$ deg N in $10^9\,\mathrm{cm}^{-2}$.

which is also present in the WACCM data. For a better comparison of the data Fig. 7 shows the
differences of the centroid altitudes to the ensemble mean, when the WACCM centroid altitude is
shifted 2 km upwards, which results in an nearly optimal agreement for most latitudes and times of
WACCM with GOMOS, SCIAMACHY and the descending leg of OSIRIS when only a global shift
between these datasets is considered. For the ascending leg of OSIRIS the optimal shift is around





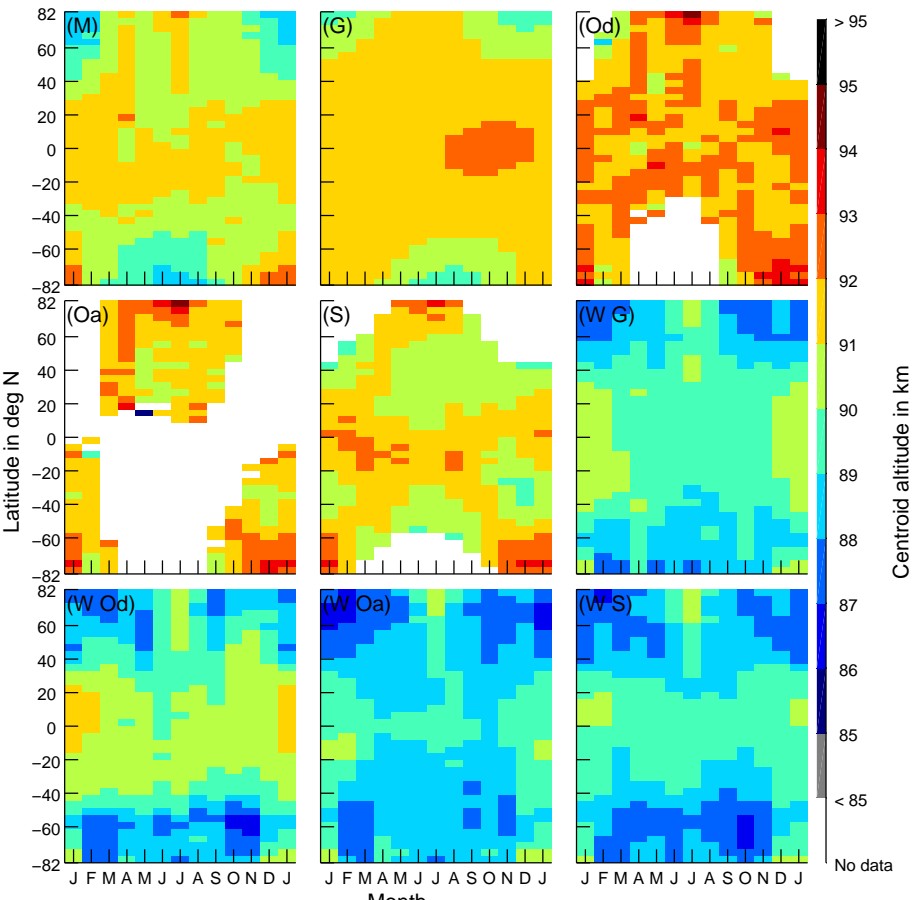

**Fig. 5.** 2008-2012 multiannual monthly mean Na layer centroid altitude from different instruments and models. (M) ensemble mean, (G) GOMOS, (Od) OSIRIS descending leg, (Oa) OSIRIS ascending leg, (S) SCIA-MACHY dayglow, (W G) WACCM colocated to the local time of ascending leg of GOMOS, (W Od) WACCM colocated to the local time of the descending leg of OSIRIS, (W Oa) WACCM colocated to the local time of the ascending leg of OSIRIS, (W S) WACCM colocated to the dayglow measurements of SCIAMACHY and the descending leg of GOMOS.

3 km.

**5   Profile widths and differences**

Figure 8 shows the FWHM of the different datasets and Fig. 9 shows the differences to the ensemble mean, except the upper left panel, which shows the ensemble mean in the same color bar as for Fig. 8, for the same reason as discussed in Sect. 3.   The FWHM is determined by finding the 50% altitudes via interpolation from the sampled grid, with a stepsize of 2 km, and taking the difference





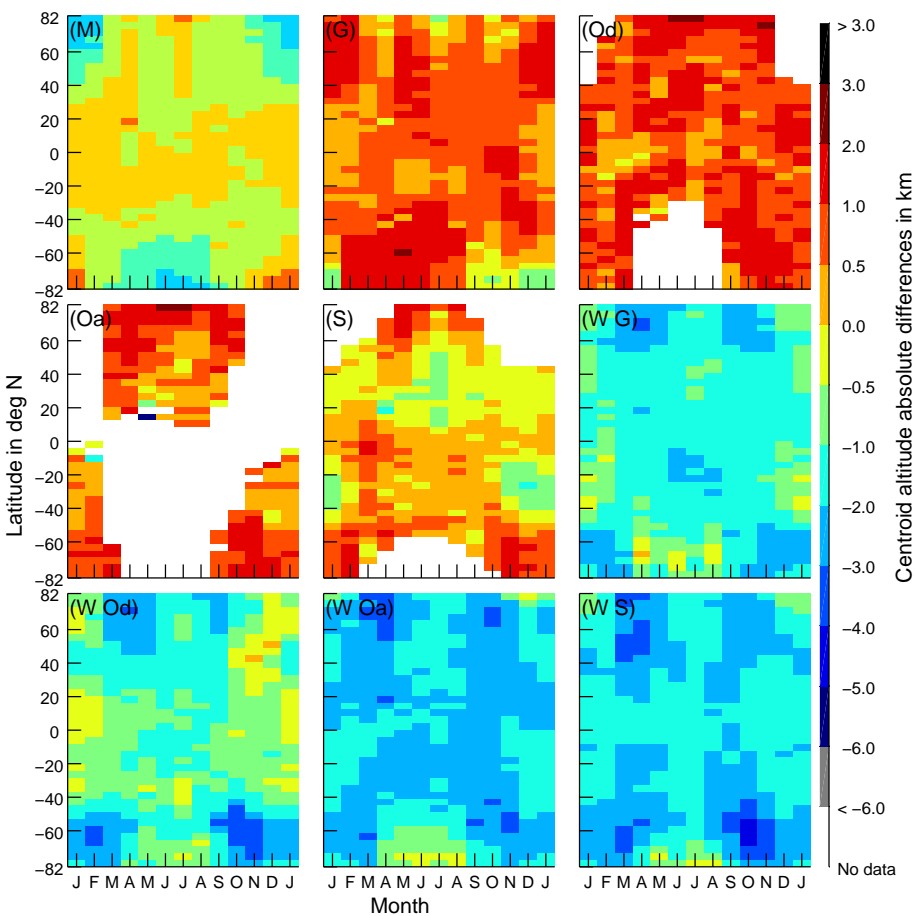

**Fig. 6.** Absolute centroid altitude differences of the individual instruments/model results to the ensemble mean. The panels are for the same results as in Fig. 5. The upper left panel shows the ensemble mean with the same color bar as in Fig. 5.

of the upper 50% altitude and the lower 50% altitude. For the GOMOS model the mean width for all GOMOS measurements of 12.2 km is used. The FWHM ranges from the sampling and resolution limit of 2 km up to 18 km. For most latitudes the FWHM is between 10 km to 16 km. For the datasets, in which the FWHM is determined to be latitude specific, the lowest profile width is observed in the polar summer. The WACCM model shows the largest profile widths in polar winter, which is not

covered by the instruments. The local time differences between the descending leg of OSIRIS and SCIAMACHY, with OSIRIS showing, e.g., slightly larger profile widths in the low latitudes than the SCIAMACHY data, are also present in the WACCM data.





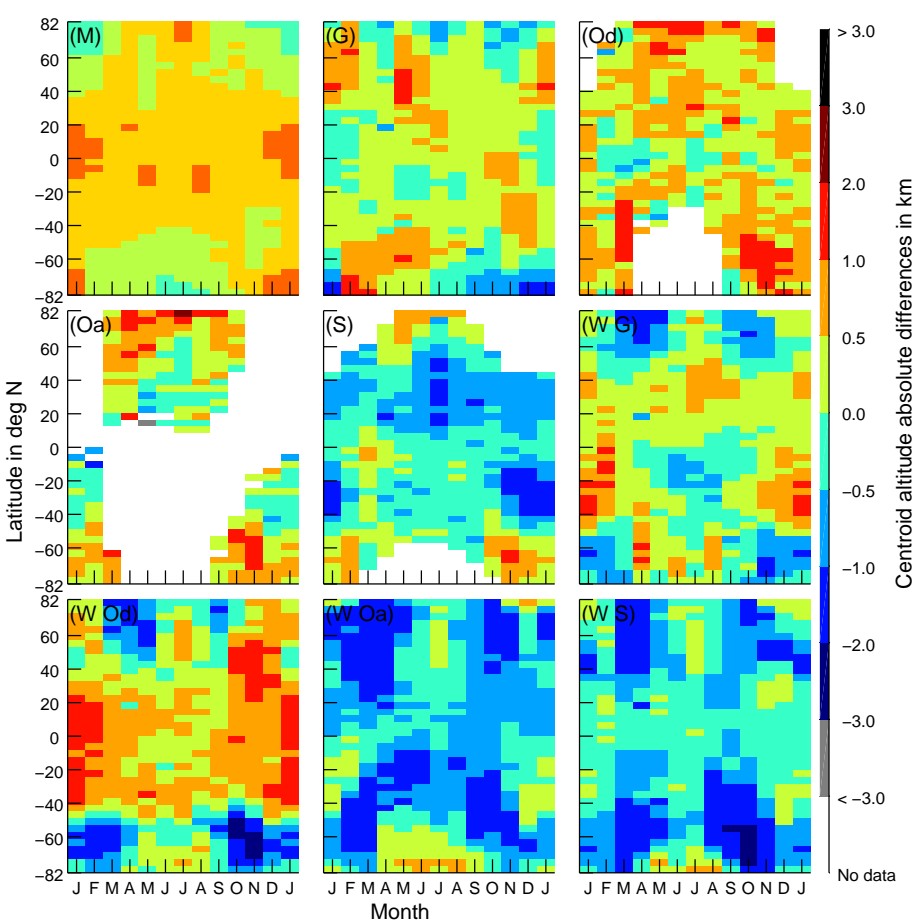

**Fig. 7.** Absolute centroid altitude differences of the individual instruments/model results to the ensemble mean with WACCM centroid altitudes shifted 2 km upwards. The panels are for the same results as in Fig. 5. The upper left panel shows the ensemble mean with the same color bar as in Fig. 5.

## 6   Conclusions

The currently available global experimental and model datasets of upper atmospheric Na densi-

ties were compared in this paper, focusing particularly on the VCDs, centroid altitudes and profile widths. Overall, there is agreement of the datasets for the VCDs with larger discrepancies for measurements carried out near the terminator. The GOMOS dataset appears to be shifted by around a month ahead of the other datasets. The VCDs vary from $0.5 \times 10^9\,\mathrm{cm}^{-2}$ to $7 \times 10^9\,\mathrm{cm}^{-2}$ near the poles and around $3 \times 10^9\,\mathrm{cm}^{-2}$ to $4 \times 10^9\,\mathrm{cm}^{-2}$ at the equator. The absolute differences of the VCD

are below $\pm 1 \times 10^9\,\mathrm{cm}^{-2}$ for most latitudes and times and exceed $\pm 2 \times 10^9\,\mathrm{cm}^{-2}$ only for very few elements of the density fields. The centroid altitudes of the different measurements are in good agreement and vary from 89 to 95 km. In the polar region the centroid altitudes are highest in the



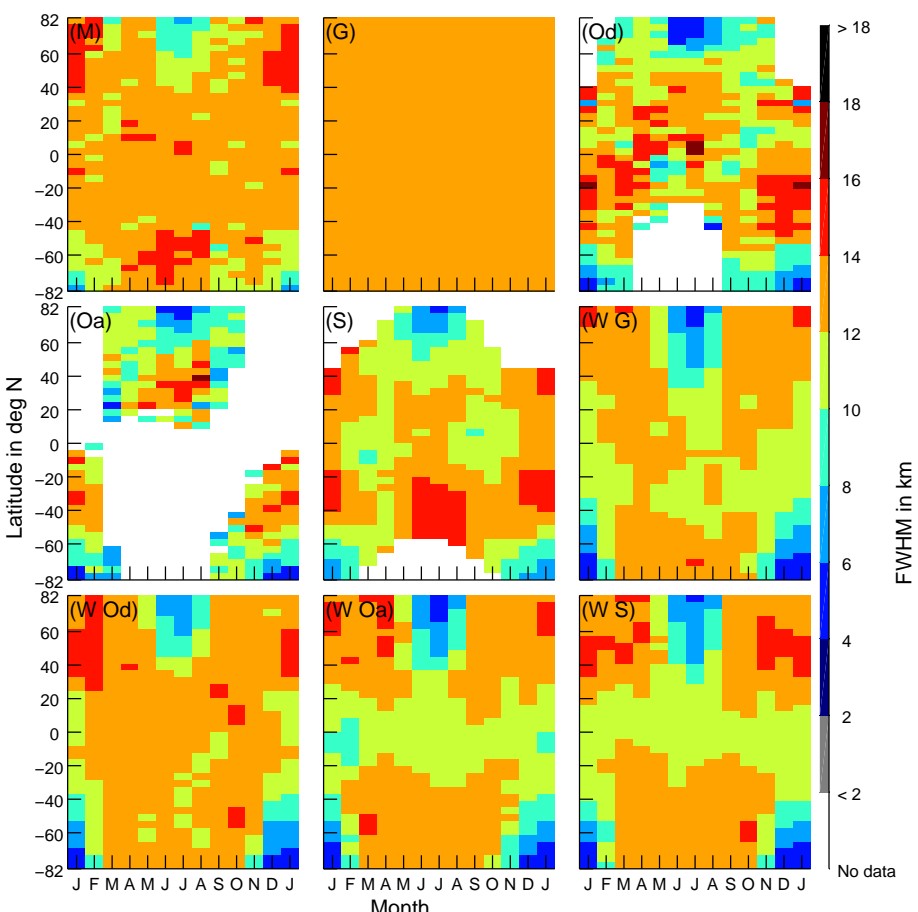

**Fig. 8.** 2008-2012 multiannual monthly mean Na layer full width at half maximum from different instruments and models. (M) ensemble mean, (G) GOMOS, (Od) OSIRIS descending leg, (Oa) OSIRIS ascending leg, (S) SCIAMACHY dayglow, (W G) WACCM colocated to the local time of ascending leg of GOMOS, (W Od) WACCM colocated to the local time of the descending leg of OSIRIS, (W Oa) WACCM colocated to the local time of the ascending leg of OSIRIS, (W S) WACCM colocated to the dayglow measurements of SCIAMACHY and the descending leg of GOMOS.

summer. The centroid altitudes of the WACCM model are systematically 2 to 4 km lower than those of the measurements. However, the local time variations between the different satellite measure-
ments are also present in the WACCM data. The FWHMs of the different datasets are in agreement and the WACCM model reproduces the local time differences between OSIRIS and SCIAMACHY well. The FWHM is around 10 to 16 km for most latitudes and times, however in the polar summer, there is a thinning out of the Na layer with low FWHM of around 5 km.

*Acknowledgements.* This work was in part supported by the European Space Agency (ESA) through the MesosphEO



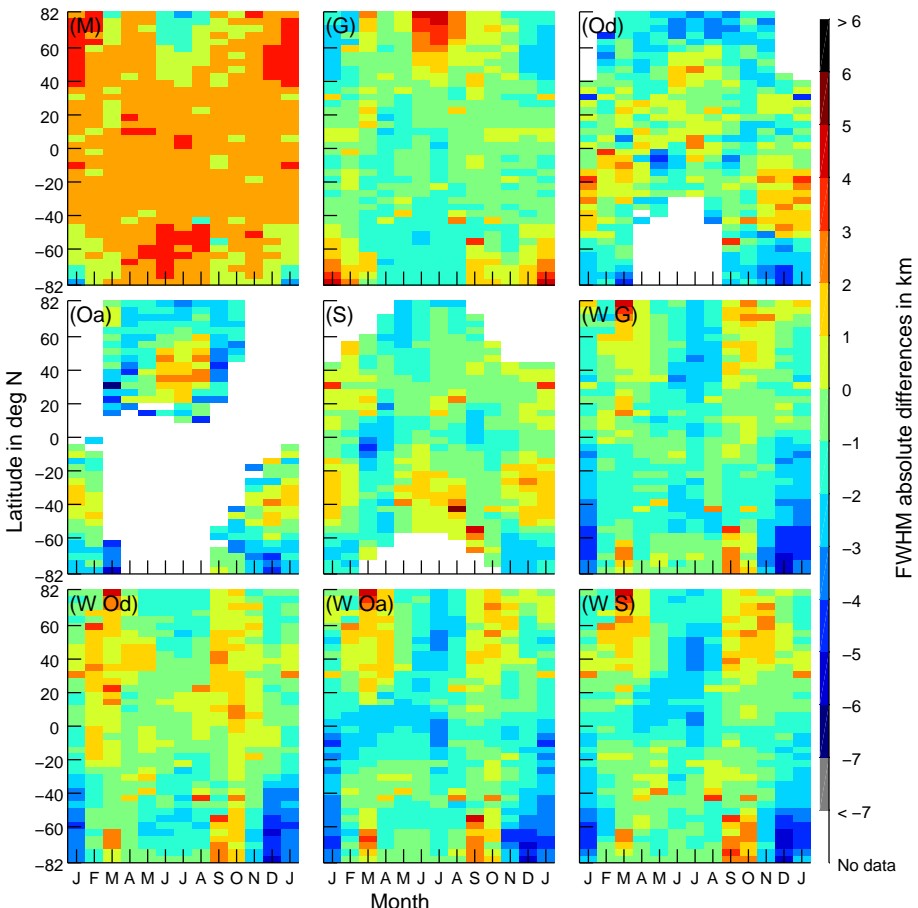

**Fig. 9.** Absolute full width at half maximum differences of the individual instruments/model results to the ensemble mean. The panels are for the same results as in Fig. 8. The upper left panel shows the ensemble mean with the same color bar as in Fig. 8.

project, by Ernst-Moritz-Arndt-University of Greifswald and the University of Bremen. SCIAMACHY is

jointly funded by Germany, the Netherlands and Belgium. We are indebted to ESA for providing the SCIA-

MACHY Level 1 data used in this study. The WACCM-Na model work was supported by the European

Reasearch Council (Project Number 291332, CODITA). We would like to thank Dr. Diego Janches at NASA

Goddard Space Flight Center for providing the Na meteoric input function data used in WACCM-Na. The

National Center for Atmospheric Research (NCAR) is sponsored by the US National Science Foundation.





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
