# Peer review of "Comparison of Global Datasets of Sodium Densities in the Mesosphere and Lower Thermosphere from GOMOS, SCIAMACHY and OSIRIS Measurements and WACCM Model Simulations from 2008 to 2012"

_Atmospheric Measurement Techniques, 2016_

## Referee Comment (RC1) · Anonymous Referee #1 · 7 Feb 2017

Review of " Comparison of Global Datasets of Sodium Densities in the Mesosphere and Lower Thermosphere from GOMOS, SCIAMACHY and OSIRIS Measurements and WACCM Model Simulations from 2008 to 2012" by Langowski et al.

**General Comments**

1. This paper describes comparisons of Na layer characteristics form observations and models, and will be useful to anyone who is using these data. While the manuscript offers an accurate and detailed description of the results, it misses many opportunities to discuss the atmospheric (or extraterrestrial) processes that may be underlying the results. Adding such discussion may shed light on the model – measurement differences, and certainly will help in keeping the reader engaged.

2. All of the results are all presented as color cross sections. While these convey a great deal of information in a small space,  I would find it useful to see a few simple line plots that focus on areas of interest. For example, all of the seasonal action is in the polar regions, and it would be instructive to see line plots, for example, of the VCD at 65S and 65N, for all of the satellites and models. There is space in the paper and it would more effectively illustrate the agreement and differences, as well as the nature of the Na layer.

3. The paper is generally well written and enjoyable to read, however, there are far too many instances of incorrect grammar. Most of the specific comments below deal with grammatical issues. Please revise the paper with an eye for correct grammar, and go beyond the comments made below.

**Specific Comments**

line 3:  You cannot use these acronyms before you define them.

lines 33-35 (and elsewhere):  Stick with either g or μg,  but not both.

line 39: Give the typical value in km (in parenthesis).

line 42; Delete "so called".

lines 43-44: You should cite the 2 papers which used observations to demonstrate that NLC particles indeed contain meteoric smoke: Havnes and Næsheim [2007] and Hervig et al. [2012].

line 45: Murphy et al. [1998] used observations to characterize smoke contained within stratospheric aerosols. This reference is relevant and should be cited as well.

line 52: Carillo–Sanchez et al. [2016] contains the current best estimate for meteoric influx (43 t/d), and you should cite that work here.

line 57: It is the velocity rather than the cost which is the principal challenge in using rockets.

lines 76, 82, and elsewhere: Phrases like ground-based, space-borne, on-board, Gaussian-shaped, etc… are not put in parenthesis.

line 93:  You cannot use these acronyms before you define them.

line 95" "good" should be "well", then "as well as" should be "in addition to"

line 101: Do not capitalize full width half maximum, it is not a proper noun.

line 104: This is an awkward sentence, please re-write.

line 108: …datasets is provided…

line 118: Delete "where the satellite flies from north to south", as this is implied by the term descending.

line 124: Do you mean "from 2003 to present" ?

lines 125-127: You can delete these sentences, as the information is fundamental to the technique, and is described elsewhere.

line 132: Do not capitalize "differential optical absorption spectroscopy". Also, since you never use the acronym DOAS, you do not need to define it.

line 139-140: This is poorly stated. You do not have equations 8, 9, and 11 in your paper. You should rather state that "…below we reproduce the expressions of Fusen et al (Equations 8, 9, and 11 therein)…"

line 139: The representation of GOMOS results as given in Fussen et al. would commonly be referred to as a climatology. Later in the paper you refer to the "GOMOS model data", which is confusing. I recommend introducing it as a climatology, and then later in the paper refer to it simply as "GOMOS" or as "the GOMOS climatology".

line 141 and elsewhere: However should have a comma before and after when it is used for emphasis.

lines 162-163: The relevance of this transformation should be explained.

line 165: SCIAMACHY should be in parentheses.

line 178: "multi-annual"

line 179: Step size in which dimension? Perhaps say "…vertical step size…".

line 180: "step size"

lines 181-183: The sentence in parentheses is not stated very well. Please re-word, and possibly take it out of parentheses.

line 184: The vertical resolution is not determined by the step size, but rather is due to the instrument field of view. Other effects may come into play, like response time, the detector sampling frequency, and the scan rate. The preceding discussion of the step size of 3.3 km, and retrieval spacing of 1 km is rather confusing. One would typically not retrieve on a finer grid than was observed. You should re-word this to clearly summarize the main points: 1) The vertical resolution of the measurements determined by the FOV, 2) the vertical spacing of the measurements due to the detector readout cadence combined with the scan rate, and 3) the spacing of the data used here. Alternately, you can get away with stating only the barest relevant numbers and give a reference for more detail.

line 198: "…to be less than 1 km."

line 214: What does the pixel resolution of 0.4 nm correspond to in regards to the measurements? Is it spatial, spectral, or?

lines 230-233: You use the acronym WACCM before you define it.

line 245: Delete "termed as".

line 247: How does the Na input function of Marsh et al compare to that given in Carillo–Sanchez et al. [2016]?

line 251: try "…daily at midnight UT during…"

Somewhere in the paper you should define the acronym for local solar time (LT), as it will be used throughout.

line 256: This paragraph is difficult to read. I think the main points could be stated very simply. Please clarify this section.

Fig. 1: The caption is poorly worded. What do you mean by "The satellite data is collocated…" ? I think this is a plot of model Na VCD results, and that the curves illustrate the satellite LT vs. latitude. Can it be stated this simply?

line 277: Missing tab. Figure 2 shows model results, not measurements. Thus this sentience is misleading.

line 278: delete "negligibly", also, can you provide a value?

line 279: You discuss the ensemble mean before it is defined. The discussion starting on line 283 should come before you present the results.

Fig. 2 (and other figures): In the description of the results you should place the notation after the description, e.g. (2dn sentence): "Results are for the ensemble mean (M), GOMOS (G), …"

line 283: Integers less than 10 are spelled out, e.g. "four",

line 292: Delete "in numbers"

line 294: Within the following paragraph, you need to occasionally refer to a specific figure (in parentheses) where the reader can see what you are describing.

All Figures: You need to label the panels sequentially by letter (e.g., a), b), etc…), on each panel itself, and then describe the figure as such in the caption. This is the standard in journals, and it will facilitate your discussion when you refer to a specific figure / panel.

line 306: "model data" should be "climatology"

line 315 (and elsewhere): "89km up to 95km" should be "89 to 95 km"

line 318: Do not capitalize mesopause.

line 320: What is "…the thermal structure in its altitude"? Please clarify this statement.

line 321: Can you comment on why the layer is higher in summer? Is it simply because of upwelling? It would be a welcome diversion to occasionally comment on the atmospheric processes behind some of the features in the measurements and models.

line 322: This sentence is unclear.

lines 329-333: This sentence is too long.

line 338: You do not need to describe how FWHM is determined, it is a fundamental that we all understand.

line 340 – 341: "model" should be "climatology". Try this: "…mean width of 12.2 km is used at all times and latitudes."

line2 341-342: What does this refer to? GOMOS? the model?

line 342:  What do you mean by "datasets"?   Is this the observations but not the model?

line 357: "regions"

line 358:  It is not the "…centroid altitudes of the WACCM model" but rather "…the modeled Na layer centroid altitudes"

---

## Referee Comment (RC2) · Anonymous Referee #2 · 8 Feb 2017

The authors compare global measurements of the mesospheric Na layer made by the satellite instruments GOMOS, SCIAMACHY and OSIRIS/Odin and modeled by WACCM-Na. The results show that the various instruments and model generally agree on the dominant geographic and seasonal variations of the Na abundance, centroid height and layer width. The most significant descrepancy is the centroid altitude as modeled by WACCM-Na which is 2-4 km lower than the satellite observations. But this issue and its probably cause had been pointed out previously by several of the

authors when they compared WACCM-Na with lidar measurements. The paper is reasonably well-written and adequately referenced. However, the satellite measurements and WACCM-Na predictions are generally consistent with ground based lidar observations of Na made at a wide variety of latitudes, including polar latitudes. Thus the results, while interesting from the perspective of confirming that the satellite measurements are consistent with each other, do not reveal any significant new science. Even so, I think this comparison is worthy of publication AMT.

---

## Short Comment (SC1) · 17 Mar 2017

Dear colleagues

In the specific comment on the manuscript's line 315, Reviewer 1 asks the authors to change the formatting of ranges of quantities. The formatting of SI units and quantities is specified in Bureau International des Poids et Mesures (2006, Sect. 5.3.3., p. 133), see also Thompson and Taylor (2008, p. v). I recommend that the authors keep the

current formatting of quantities, because it agrees with the cited references.

Kind regards
Tim Dunker

**References**

Bureau International des Poids et Mesures: SI Brochure: The International System of Units (SI), 8th edition, updated in 2014, http://www.bipm.org/utils/common/pdf/si_brochure_8.pdf (accessed: 17 March 2017), 2006.

Thompson, A. and Taylor, B. N.: Guide for the Use of the International System of Units (SI), NIST Special Publication 811 (SP 811), National Institute of Standards and Technology, http://physics.nist.gov/cuu/pdf/sp811.pdf (accessed: 17 March 2017), 2008.

---

## Author Comment (AC1) · 13 Apr 2017

**Response to the Reports of anonymous Referee #1 and #2 of the manuscript:**
**"Comparison of Global Datasets of Sodium Densities in the Mesosphere and**
**Lower Thermosphere from GOMOS, SCIAMACHY and OSIRIS Measurements and**
**WACCM Model Simulations from 2008 to 2012" by Langowski et al.**

We would like to thank the Referees for the critical reading and especially referee #1 for the many recommendations and ideas to improve this manuscript. For a detailed answer we repeat the reports in black letters and will add answers to remarks in blue.

**Response to Referee #1:**

Review of "Comparison of Global Datasets of Sodium Densities in the Mesosphere and Lower Thermosphere from GOMOS, SCIAMACHY and OSIRIS Measurements and WACCM Model Simulations from 2008 to 2012" by Langowski et al.

**General Comments**

1. This paper describes comparisons of Na layer characteristics form observations and models, and will be useful to anyone who is using these data. While the manuscript offers an accurate and detailed description of the results, it misses many opportunities to discuss the atmospheric (or extraterrestrial) processes that may be underlying the results. Adding such discussion may shed light on the model – measurement differences, and certainly will help in keeping the reader engaged.

The paper is focused on a comparison of measurement techniques, and using a global model to link them together, but is not concerned with the underlying science issues that are dealt with in the references provided.
We added a sentence in the introductory section to make this more clear.

2. All of the results are all presented as color cross sections. While these convey a great deal of information in a small space, I would find it useful to see a few simple line plots that focus on areas of interest. For example, all of the seasonal action is in the polar regions, and it would be instructive to see line plots, for example, of the VCD at 65S and 65N, for all of the satellites and models. There is space in the paper and it would more effectively illustrate the agreement and differences, as well as the nature of the Na layer.

We agree with the Dilemma of information density and readability and added a new plot and section showing and discussing the 3 discussed key characteristics of the Na layer at around 65N/S.

3. The paper is generally well written and enjoyable to read, however, there are far too many instances of incorrect grammar. Most of the specific comments below deal with grammatical issues. Please revise the paper with an eye for correct grammar, and go beyond the comments made below.

We are sorry, that far too many instances of incorrect grammar occurred. The manuscript has been proofread by several native speakers. We tried our best to correct all mentioned mistakes and furthermore improved the grammar of the manuscript.

**Specific Comments**

line 3: You cannot use these acronyms before you define them.
We placed the definitions at the positions of the first appearance in abstract and text.

lines 33-35 (and elsewhere): Stick with either g or µg, but not both.

changed both to kg

line 39: Give the typical value in km (in parenthesis).

corrected as suggested

line 42; Delete "so called".

corrected as suggested

lines 43-44: You should cite the 2 papers which used observations to demonstrate that
NLC particles indeed contain meteoric smoke: Havnes and Næsheim [2007] and Hervig
et al. [2012].

Cited these papers

line 45: Murphy et al. [1998] used observations to characterize smoke contained within
stratospheric aerosols. This reference is relevant and should be cited as well.

Cited this paper

line 52: Carillo–Sanchez et al. [2016] contains the current best estimate for meteoric
influx (43 t/d), and you should cite that work here.

Cited this paper

line 57: It is the velocity rather than the cost which is the principal challenge in using
rockets.

I did not fully understand why velocity is more important here.
This statement was meant on a global perspective and a per profile cost estimation, considering e.g.,
satellites are  able to take a daily profile on global positions for several years.
For the investigation of singular events rockets are of course cheaper and better suited than
satellites, e.g., because they can be started at the exact time of an interesting atmospheric event.
We changed the sentence to a per profile cost statement.

lines 76, 82, and elsewhere: Phrases like ground-based, space-borne, on-board, Gaussian-
shaped, etc… are not put in parenthesis.

We actually used inverted commas and the idea came from native speakers, but we changed that.

line 93: You cannot use these acronyms before you define them.

As stated above, now the acronyms are mentioned earlier.

line 95" "good" should be "well", then "as well as" should be "in addition to"

corrected as suggested

line 101: Do not capitalize full width half maximum, it is not a proper noun.

FWHM is a widely accepted abbreviation and is e.g. found in Wikipedia and in most of the first
pages of a google search for "fwhm". It is furthermore an acronym and, therefore, can be spelled in
capital letters. I previously used this abbreviation in other peer reviewed publications and it was not
considered to be wrong. It might be technical wrong but it is conventionally used.

line 104: This is an awkward sentence, please re-write.

We tried to reformulate the sentence, see also next point

line 108: …datasets is provided…

I reread about the word information, and although it is not a singular word, as (e.g. it is not allowed
to use say an information or one information) it is conjugated like a singular word. I checked the
manuscript for every use of the word.

line 118: Delete "where the satellite flies from north to south", as this is implied by the
term descending.

Descending is describing something to go downward. This is actually not intuitive for the motion of
an object in the Earth atmosphere, as downward is connected with a negative change of altitude.
This is clearly not the case and what is meant is a downward movement of the satellite on a 2
dimensional projection of Earth with the north on the top. When I'm in Berlin and going to Munich,
I'm going southward and not downwards. Therefore, I would like to leave this as it is.

line 124: Do you mean "from 2003 to present" ?

The intention of this statement was to make clear, that from the satellite experiments that measure

Na on a global scale for long time periods, the first Na data products where presented in Fussen 2004, which used data for the year 2003.

lines 125-127: You can delete these sentences, as the information is fundamental to the technique, and is described elsewhere.

This is actually important for the dataset, as it is a mix of ascending and descending node measurements, with different weighting factors, so that it is hard to interpret the local time effects, as it is not always clear which local time has the dominant weighting.

line 132: Do not capitalize "differential optical absorption spectroscopy". Also, since you never use the acronym DOAS, you do not need to define it.

I agree, that this acronym is used only once in the paper. In the literature sometimes only the word DOAS is used without a further explanation. As I would like to give readers with differing level of background knowledge the opportunity to understand what is meant, I would like to leave it as it is.

line 139-140: This is poorly stated. You do not have equations 8, 9, and 11 in your paper. You should rather state that "…below we reproduce the expressions of Fusen et al (Equations 8, 9, and 11 therein)…"

This sounds like a good correction, so we changed that.

line 139: The representation of GOMOS results as given in Fussen et al. would commonly be referred to as a climatology. Later in the paper you refer to the "GOMOS model data", which is confusing. I recommend introducing it as a climatology, and then later in the paper refer to it simply as "GOMOS" or as "the GOMOS climatology".

As suggested, we introduced the word GOMOS climatology and used this word later instead of 'model formulae'.This has now been corrected.

line 141 and elsewhere: However should have a comma before and after when it is used for emphasis.

The removal of those commas was suggested by an American native speaker during the proofreading, who found that there were to many stops in the sentences. Nevertheless, online dictionaries suggest this rule too (comma before and after for A.E. and after for B.E.), so we follow the referee's recommendations for however and words where the same rule applies ( cunjunctive adverbs ... however, furthermore, therefore etc.).

lines 162-163: The relevance of this transformation should be explained.

In the interpretation of the Formula by Fussen 2010, I made a small mistake, and interpreted zeta in formula (11) as the FWHM, which is incorrect. Therefore, sigma also was different. The conversion between zeta and FWHM is sqrt(2pi)/sqrt(8ln2), so that the FWHM is 6% smaller and only 11.46 km. Fortunately, this does not qualitatively affect the discussion in the paper. This mistake was corrected throughout the manuscript in text and all affected figures.

line 165: SCIAMACHY should be in parentheses.

corrected as suggested

line 178: "multi-annual"

corrected as suggested (everywhere in text)

line 179: Step size in which dimension? Perhaps say "…vertical step size…".

corrected as suggested

line 180: "step size"

corrected as suggested (everywhere in text)

lines 181-183: The sentence in parentheses is not stated very well. Please re-word, and possibly take it out of parentheses.

We removed this sentence, (see next point)

line 184: The vertical resolution is not determined by the step size, but rather is due to the instrument field of view. Other effects may come into play, like response time, the detector sampling frequency, and the scan rate. The preceding discussion of the step size of 3.3 km, and retrieval spacing of 1 km is rather confusing. One would typically not retrieve on a finer grid than was observed. You should re-word this to clearly summarize the main points: 1) The vertical resolution of the measurements determined by the FOV,

2) the vertical spacing of the measurements due to the detector readout cadence combined with the scan rate, and 3) the spacing of the data used here. Alternately, you can get away with stating only the barest relevant numbers and give a reference for more detail.

These are good points for the discussion of the vertical resolution.
The scanning sequence of a SCIAMACHY limb scan is well described here:
 http://atmos.caf.dlr.de/projects/scops/sciamachy_book/sciamachy_book_ch4.pdf , page52
The seqence consists of a short upward movement of the IFoV and a longer horizontal scan.
The Instantaneous FoV at the tangent altitude has a vertical extent  of 2.6 km and the maximum vertical extent of the FOV is 3.6 km.
Roscoe and Hill (2001) ("Vertical resolution of oversampled limb-sounding measurements from satellites and aircraft") showed, that for typical satellite FOV (which is unfortunately not further quantified) the vertical resolution is 3-5 km and the SCIAMACHY ozone retrievals, which include a FOV integration show averaging kernels with a FWHM of about 4.5 km.

Additional reduction of the vertical resolution can also come from the radiative transfer model, i.e. the regularization. Here, we use vertical constraints that are weak enough, to not further smooth the vertical resolution, so that we assume the vertical resolution to be close to the spacing of the data and small enough compared to the width of the Na layer that smoothing effects on the profile are only insignificant.
Maybe I'm using the wrong term, but retrieving on a finer spacing as the instruments resolution is done, because the numerical errors become smaller. But, of course, the vertical resolution can not not get better doing this.

We decided to significantly shorten the paragraph, as its complexity to importance ratio is rather high.

line 198: "…to be less than 1 km."
Corrected as suggested
line 214: What does the pixel resolution of 0.4 nm correspond to in regards to the measurements? Is it spatial, spectral, or?
We mean the spectral sampling here, and corrected this in the manuscript.
lines 230-233: You use the acronym WACCM before you define it.
Corrected as suggested
line 245: Delete "termed as".
Corrected as suggested
line 247: How does the Na input function of Marsh et al compare to that given in Carillo–Sanchez et al. [2016]?
The annual and global mean of Na input used in this paper is identical as in Marsh et al. (2003), which is about 2027.47 atoms cm-2 s-1, about 8.34 times smaller than Carillo-Sanchez et al. (2016) (16905 atoms cm-2 s-1). Overall, the Na injection rate in Carillo-Sanchez et al. (2016) is much larger with maximum value of 0.012  atom cm-3 s-1 around 87 km than in Marsh et al. (2003) which has a maximum value of 0.002 atom cm-3 s-1 around 105 km. This appears to be because the horizontal resolution of WACCM is too coarse to resolve gravity waves with wavelengths smaller than ~150 km, so that vertival wave-driven chemical transport is not fully modeled, requiring the Na input to be reduced in the model.

line 251: try "…daily at midnight UT during…"
Corrected as suggested
Somewhere in the paper you should define the acronym for local solar time (LT), as it will be used throughout.
Defined it and used it

line 256: This paragraph is difficult to read. I think the main points could be stated very simply. Please clarify this section.
We reformulated the paragraph.
Fig. 1: The caption is poorly worded. What do you mean by "The satellite data is collocated…" ? I think this is a plot of model Na VCD results, and that the curves illustrate the satellite LT vs. latitude. Can it be stated this simply?
We tried to clarify the caption. As model data is available for all local times, we only use the model data within a certain time difference from the satellites LT, and discard the remaining model data before forming the zonal averages, colocated to each instrument.

line 277: Missing tab.
There is no indent for the first paragraph of a section. We use the standard latex template for the journal, so we assume it is meant this way.
Figure 2 shows model results, not measurements. Thus this sentience is misleading.
I do not understand this comment. Figure 2 shows VCDs of all used instruments and models and not just model results. Only Figure 1 shows model results only.
line 278: delete "negligibly", also, can you provide a value?
We can provide a rough highest limit. Which, however, is probably too high for most of the altitudes above 105 km.
For GOMOS we use a Gaussian profile, OSIRIS data is only available between 76 and 106 km and the WACCM model (see figure below) shows densities below 1 cm^-3 below 70km altitude and quickly falling density at the upper edge.

[Figure]

For SCIAMACHY, figure 19 in Langowski et al. 2016 shows a typical Na profile with errors. The densities above 105 km are below 100 atom per cubic centimeter, which is lower than the error of the density there (note that Na densities are forced to be positive, so that the positive density there is assumed to be purely coming from noise.).

There are reasons from models and measurements perspective, that we don't expect a significant amount of Na above and below this vertical region. At the lower boundary the O3 densities are too high, which lead to fast reactions into molecular species.

At the upper boundary the injection of ablated metals is limited. The atmospheric density decreases exponentially, so that above a certain altitude no strong heating and therefore ablation is taking place. From the radiative transfer model point of view, a large column of Na above 150 km would lead to a strong peak of the retrieved density at the upper most retrieval altitude, which is not observed.

SCIAMACHY measures up to 150 km, but using the values between 105 and 150 km would include large uncertainties.

For example, a $50cm^{-3}$ density between 105 and 150 km would result in a VCD of ~0.25E9 $cm^{-2}$, with a similarly large error. Therefore, considering these altitudes would reduce the accuracy of the here shown results, as the algorithms are tuned (e.g. regularization, considering of self absorption) to be exact in the region of maximum density.

Note, that for a better estimation of the densities between 105 and 150 km stronger averaging of the data would lead to more exact results, but would result in a worse temporal resolution.

In the manuscript I now use the value below 100 $cm^{-3}$, which, however, is a rather high upper limit.

line 279: You discuss the ensemble mean before it is defined. The discussion starting on

Moved the paragraph for the ensemble mean to the front.

line 283 should come before you present the results.

Moved the paragraph for the ensemble mean to the front.

Fig. 2 (and other figures): In the description of the results you should place the notation after the description, e.g. (2dn sentence): "Results are for the ensemble mean (M), GOMOS (G), …"

Corrected as suggested

line 283: Integers less than 10 are spelled out, e.g. "four",

It's actually clear what is meant, but I changed it.

line 292: Delete "in numbers"

Corrected as suggested

line 294: Within the following paragraph, you need to occasionally refer to a specific figure (in parentheses) where the reader can see what you are describing.

Here figure 2  shows the VCDs which similarities and differences are discussed, as a help figure 3 and figure 4 show absolute and relative differences of the individual datasets to the ensemble mean. When comparing the data I would recommend looking at all of these 3 figures. As, e.g., timeshifts result in larger absolute and relative errors, even if the visual agreement in the actual data does not look so bad. There are no other figures mentioned in this section and all three figures deal with the topic VCD. So I don't see a reason, why it shouldn't be clear that only the figures mentioned in the section are discussed here.

Each dataset has one panel in each of the plots, and I don't see an improvement from e.g. GOMOS Na VCD compared to GOMOS Na VCD, shown in Fig. 2 panel GOMOS, this was double information, which would irritate the reading flow.

All Figures: You need to label the panels sequentially by letter (e.g., a), b), etc…), on each panel itself, and then describe the figure as such in the caption. This is the standard in journals, and it will facilitate your discussion when you refer to a specific figure / panel.

I think the labeling is more confusing. This would make an a,b,c,d,e.... labeling for 8 figures, where

the reader had to read what a,b,c,d,e... is everytime. However, the figures are structured the way, that the same instrument, model data set is represented in the same panel in all of the 8 figures. This is clarified by using the same labeling for each panel in these 8 figures.

line 306: "model data" should be "climatology"

Corrected as suggested

line 315 (and elsewhere): "89km up to 95km" should be "89 to 95 km"

Corrected as suggested

line 318: Do not capitalize mesopause.

Corrected as suggested

line 320: What is "…the thermal structure in its altitude"? Please clarify this statement.

This means, that the altitude of the WACCM model is lower, because the temperature profile in the mesopause region is also lower and the model is very sensitive to this.

line 321: Can you comment on why the layer is higher in summer? Is it simply because of upwelling? It would be a welcome diversion to occasionally comment on the atmospheric processes behind some of the features in the measurements and models.

The atmospheric processes behind most of the features are partly discussed in the individual papers about the compared datasets.

Here, I would like to stick just to the comparison of the data itself.

We do not think the process here is upwelling. In the polar summer the mesopause is lower and as the circulation pattern changes above the mesopause, there should be a balance at 90 km.

We think it is a thinning out of the layer at both edges, at the top due to more ionisation into Na+ and on the bottom due either a) lower mesopause temperatures leading to a stronger reaction into the NaHCO3 or b) uptake of Na into NLCs. (In WACCM only b) is not included, so the effect comes from a) there).

line 322: This sentence is unclear.

The polar winter is not covered by all the satellite instruments. However, the largest seasonal variations are found at high latitudes. Therefore, we look at compromise latitudes, which are high, but include at least spring and autumn.

At 60°N, the Na centroid altitude retrieved with SCIAMACHY is about 2 km higher in June compared to March and September. For OSIRIS descending leg the altitude at 82N is 90-91km in March and November and above 94 km in July.

For GOMOS the altitude in January, at 82N is 89-90 km and in summer 91 to 92 km. For WACCM in January 87-88km and in July 90-91km. So overall there is a up to 4 km higher layer in summer than in winter.

lines 329-333: This sentence is too long.

Added more periods and commas into the sentence

line 338: You do not need to describe how FWHM is determined, it is a fundamental that we all understand.

I shortened the sentence but find it important to mention the interpolation and that by that a more accurate FWHM can be calculated than with the 2 km steps, as this grid was too coarse.

line 340 – 341: "model" should be "climatology". Try this: "…mean width of 12.2 km is used at all times and latitudes."

Corrected as suggested

line2 341-342: What does this refer to? GOMOS? the model?

This is equation (11) from Fussen et al 2010. For the climatology the mean of all measurements is used, at all times and latitudes.

line 342: What do you mean by "datasets"? Is this the observations but not the model?

No, it is all but GOMOS.

line 357: "regions"

Corrected as suggested

line 358: It is not the "…centroid altitudes of the WACCM model" but rather "…the modeled Na layer centroid altitudes"

Corrected as suggested

**Response to Referee #2:**

The authors compare global measurements of the mesospheric Na layer made by the satellite instruments GOMOS, SCIAMACHY and OSIRIS/Odin and modeled by WACCM-Na. The results show that the various instruments and model generally agree on the dominant geographic and seasonal variations of the Na abundance, centroid height and layer width. The most significant descrepancy is the centroid altitude as modeled by WACCM-Na which is 2-4 km lower than the satellite observations. But this issue and its probably cause had been pointed out previously by several of the authors when they compared WACCM-Na with lidar measurements. The paper is reasonably well-written and adequately referenced. However, the satellite measurements and WACCM-Na predictions are generally consistent with ground based lidar observations of Na made at a wide variety of latitudes, including polar latitudes. Thus the results, while interesting from the perspective of confirming that the satellite measurements are consistent with each other, do not reveal any significant new science. Even so, I think this comparison is worthy of publication AMT.

There where no suggestions for improvements.